# Biosulfidogenesis Mediates Natural Attenuation in Acidic Mine Pit Lakes

**DOI:** 10.3390/microorganisms8091275

**Published:** 2020-08-21

**Authors:** Charlotte M. van der Graaf, Javier Sánchez-España, Iñaki Yusta, Andrey Ilin, Sudarshan A. Shetty, Nicole J. Bale, Laura Villanueva, Alfons J. M. Stams, Irene Sánchez-Andrea

**Affiliations:** 1Laboratory of Microbiology, Wageningen University, Stippeneng 4, 6708 WE Wageningen, The Netherlands; sudarshanshetty9@gmail.com (S.A.S.); fons.stams@wur.nl (A.J.M.S.); 2Geochemistry and Sustainable Mining Unit, Dept of Geological Resources, Spanish Geological Survey (IGME), Calera 1, Tres Cantos, 28760 Madrid, Spain; j.sanchez@igme.es; 3Dept of Mineralogy and Petrology, University of the Basque Country (UPV/EHU), Apdo. 644, 48080 Bilbao, Spain; i.yusta@ehu.eus (I.Y.); andrey.ilin@ehu.eus (A.I.); 4NIOZ Royal Netherlands Institute for Sea Research, Department of Marine Microbiology and Biogeochemistry, and Utrecht University, Landsdiep 4, 1797 SZ ‘t Horntje, The Netherlands; nicole.bale@nioz.nl (N.J.B.); laura.villanueva@nioz.nl (L.V.); 5Centre of Biological Engineering, University of Minho, Campus de Gualtar, 4710-057 Braga, Portugal

**Keywords:** acidophiles, sulfate reduction, sulfur reduction, sulfur disproportionation, biosulfidogenesis, bioremediation, sulfide neoformation, lipid biomarker

## Abstract

Acidic pit lakes are abandoned open pit mines filled with acid mine drainage (AMD)—highly acidic, metalliferous waters that pose a severe threat to the environment and are rarely properly remediated. Here, we investigated two meromictic, oligotrophic acidic mine pit lakes in the Iberian Pyrite Belt (IPB), Filón Centro (Tharsis) (FC) and La Zarza (LZ). We observed a natural attenuation of acidity and toxic metal concentrations towards the lake bottom, which was more pronounced in FC. The detection of Cu and Zn sulfides in the monimolimnion of FC suggests precipitation of dissolved metals as metal sulfides, pointing to biogenic sulfide formation. This was supported by microbial diversity analysis via 16S rRNA gene amplicon sequencing of samples from the water column, which showed the presence of sulfidogenic microbial taxa in FC and LZ. In the monimolimnion of FC, sequences affiliated with the putative sulfate-reducing genus *Desulfomonile* were dominant (58%), whereas in the more acidic and metal-enriched LZ, elemental sulfur-reducing *Acidianus* and *Thermoplasma spp.*, and disproportionating *Desulfocapsa spp*. were more abundant. Furthermore, the detection of reads classified as methanogens and *Desulfosporosinus spp.*, although at low relative abundance, represents one of the lowest pH values (2.9 in LZ) at which these taxa have been reported, to our knowledge. Analysis of potential biomarker lipids provided evidence that high levels of phosphocholine lipids with mixed acyl/ether glycerol core structures were associated with *Desulfomonile*, while ceramide lipids were characteristic of *Microbacter* in these environments. We propose that FC and LZ function as natural bioremediation reactors where metal sulfide precipitation is mediated by biosulfidogenesis starting from elemental sulfur reduction and disproportionation at an early stage (LZ), followed by sulfate reduction at a later stage (FC).

## 1. Introduction

Acidic pit lakes form when abandoned open pit mines are allowed to flood, and exposure of the polymetallic sulfide ores to water and oxidants such as oxygen (O_2_) and ferric iron (Fe^3+^) catalyzes their oxidative dissolution. This results in the generation of acid mine drainage (AMD)—highly acidic, metalliferous waters that pose a severe threat to the environment. Due to economic and technical factors, proper remediation of acidic pit lakes rarely occurs, and these lakes persist for decades. Assessment of the long-term hazards posed by these lakes requires a better understanding of the different factors influencing their development, which include the microbial activity in these waters. In the past decade, several studies of acidic pit lakes in the Iberian Pyrite Belt (IPB), one of the largest massive sulfide districts on Earth and harboring many such lakes due to a long history of mining [1,2,3], demonstrated the tight connection between microbiology and physicochemical dynamics in IPB pit lakes [4,5,6,7], similar to what is known for AMD streams [8,9,10] and the naturally acidic, metalliferous Tinto River [11,12,13,14].

A common characteristic of acidic pit lakes in the IPB is their conical shape, and relatively small surface area and large depth [3,15]. In most cases, density differences between layers and a low surface to volume ratio result in the establishment of *meromixis*, a permanent stratification of the water column into an upper oxygenated layer (mixolimnion), and a lower anoxic layer (monimolimnion). Due to the abundance of pyrite (FeS_2_) in the IPB, AMD waters in the there are characterized by extremely high concentrations of dissolved iron and sulfate (SO_4_^2−^). The generally low organic carbon content and low phosphate availability (due to adsorption to ferric iron precipitates) makes most acidic pit lakes oligotrophic [5,16]. As a result of these extreme conditions, AMD is generally characterized by low microbial diversity, and often dominated by microorganisms involved in the oxidation and reduction of iron [17,18]. For example, a cultivation-independent study of the water column of Tinto River, in the IPB, showed that 80% of 16S rRNA gene sequences were assigned to only three genera, *Leptospirillum*, *Acidiphilium* and *Acidithiobacillus* [19], which predominantly mediate the cycling of iron and sulfur. Similarly, cultivation-independent studies of four acidic pit lakes in the IPB, Cueva de la Mora, Herrerías-Guadiana, Concepcion and Nuestra Senora del Carmen, also showed a low diversity, with most detected taxa involved in the cycling of iron [4,5,6].

In addition to the importance of microorganisms for the cycling of iron, microorganisms involved in the sulfur cycle also have a profound effect on the physicochemical dynamics in AMD waters. Sulfur-oxidizing bacteria, predominantly *Acidithiobacillus* spp., contribute to AMD formation by mediating the oxidation of the intermediate sulfur compounds formed by indirect leaching of metal sulfides, such as elemental sulfur (S_8_^0^), to SO_4_^2−^. generating further acidity [20]. In contrast, reductive microbial sulfur metabolism—the reduction of oxidized sulfur compounds to sulfide (H_2_S) (biosulfidogenesis)—has the opposite effect. The biogenic H_2_S mediates the removal of dissolved metals by their precipitation with H_2_S as metal sulfides, effectively reversing AMD formation. Furthermore, at low pH, sulfidogenesis from SO_4_^2−^ has an attenuating effect on the acidity through the consumption of protons (H^+^). For example, the complete oxidation of acetic acid to CO_2_ with SO_4_^2−^ as electron acceptor at a pH below the pK_a_ of acetic acid (4.75), H_2_S (7.02) and CO_2_ (6.37) requires H^+^ (Equation (1)). This is not the case for sulfidogenesis from S_8_^0^, as this reaction is H^+^-neutral at low pH (Equation (2)) [21].
(1)$$C_{2}H_{4}O_{2}+SO_{4}^{2-}+2H^{+}\to 2\;CO_{2}+\;H_{2}S+2\;H_{2}O$$
(2)$$C_{2}H_{4}O_{2}+4\;S_{8}^{0}+2\;H_{2}O\to 2\;CO_{2}+\;4\;H_{2}S$$

The role of acidotolerant or acidophilic sulfidogenic microorganisms, predominantly sulfate-reducing bacteria (SRB), for the bioremediation of AMD through metal sulfide precipitation is well recognized [21,22,23]. However, most SRB isolated from acidic environments such as AMD streams and the Tinto River are merely acidotolerant or moderately acidophilic, for example *Thermodesulfobium narugense* (pH 4–6.5) [24], *Thermodesulfobium acidiphilium* (pH 3.7–6.5) [25], *Desulfosporosinus acididurans* (pH 3–7) [26] and *Desulfosporosinus acidiphilus* (pH 3.6–5.5) [27]. This is likely related to the fact that in these environments, SRB were detected mostly in sediments [28], where they are potentially protected from the ‘poly-extreme’ conditions (low pH, high salinity, high metal concentrations) through the creation of micro-niches, although this is still a matter of debate [29]. Very few reports exist of SRB and biosulfidogenesis in the water column of highly acidic water bodies. In the IPB, this was until recently only reported in the acidic pit lake Cueva de la Mora [4,5,7]. In the water column, the potential for formation of microenvironments is minimal [29], and these SRB likely are more resistant to extreme conditions, making them valuable for biotechnological applications.

We recently reported a high abundance of SRB in the water column of the extremely acidic and saline pit lake formed in the Brunita open pit mine (southeast Spain), as well as their attenuating effect on the extreme physicochemical characteristics of this lake [30]. Most notable was the amelioration of acidity (pH ~2.0 in the mixolimnion vs. pH 5.0 in the lower monimolimnion) with the partial or complete removal of certain major (Cu, Zn) and trace (Cd, Cr, Pb, Th, Ni, Co) metals from the water column. The observation of abundant Cu and Zn sulfide particles in the SEM-EDS analysis of suspended particulate matter (SPM) indicated the removal of these metals as sulfide precipitates. This abundance of Cu and Zn sulfides, together with the dominance of 16S rRNA gene sequences from the SRB genus *Desulfomonile* (58.8% of total filtered reads) [31,32], supported the functioning of this pit lake as a large-scale natural bioremediation reactor, where biosulfidogenesis by acidophilic SRB mediated metal sulfide precipitation and neutralization of acidity.

Here, we study two previously microbiologically unexplored acidic mine pit lakes in the IPB, Filón Centro (FC) and La Zarza (LZ), that flooded in the 1960s and 1990s, respectively. Although metal concentrations and salinity in FC and LZ are not as extreme as in the Brunita pit lake, where they are among the highest reported for acidic pit lakes to date [3], the nutrient conditions in Brunita are more favorable than in FC and LZ due to input of nitrogen and phosphorus from its surroundings, and the resulting availability of organic carbon due to phototrophic production. This is not the case in FC and LZ, and the detection of SRB in the water column of these two pit lakes would not only widen the physicochemical limits at which biological sulfate reduction is detected, but would also have great potential for application in AMD bioremediation technologies. We hypothesized that if biosulfidogenesis has triggered the attenuation of extreme conditions in these lakes, as per Brunita pit lake, then this attenuation process will have progressed further in the older pit lake (FC). We investigated this hypothesis through combined physicochemical and microbiological characterization. For the microbiological characterization of the water column, we combined 16S rRNA gene amplicon sequencing with intact polar lipid (IPL) analysis, as certain IPL profiles could serve as biomarkers for specific taxonomic groups relevant to natural bioremediation processes in AMD environments.

## 2. Materials and Methods

### 2.1. Study Sites

FC (37°35′27.4″ N, 7°07′27.5″ W) was mined for copper and pyrite [33]. This mine was first exploited in the 19th century until 1880, and then re-opened in 1956 until its closure in 1968 [33,34]. The open pit is 600 m long and 230 m wide. The pit lake that formed after the mine flooded has a surface area of 3.57 ha, with a maximum length of 430 m, a maximum width of 140 m (Figure 1a,c), and a maximum depth of 50 m. LZ (37°42′35.5″ N, 6°51′01.4″ W) was mined for the extraction of pyrite (for sulfuric acid production), copper (chalcopyrite), and zinc (sphalerite) by both underground and open pit methods. The open pit operations started in 1886 and lasted for almost a century. In 1980, the mine was only exploited by underground mining, and the mine complex was finally closed in 1991 [35]. Some maintenance and dewatering works were conducted until the pumps were stopped in 1995, and the pit had been flooding continuously until 2015. The resulting pit lake currently has a surface area of 8 ha, with a maximum length of 700 m, a variable width of between 42 m (western side) and 200 m (eastern side), and a maximum depth of 80 m (Figure 1b,d). The total depth of the open pit is 140 m.

### 2.2. Field Site Exploration and Sample Collection

FC was sampled during seven different field campaigns in October and December 2017, October 2018, March, May and December 2019, and July 2020. LZ was only sampled in two field campaigns in October and December 2017, due to extremely difficult access for rubber boats and scientific equipment. Physicochemical profiles of the water column in both lakes were obtained using a Hydrolab MS5 multi-parametric probe (Hach, Loveland, CO, USA) calibrated according to the manufacturer’s instructions. Depth profiles were determined for pH, oxidation–reduction potential (ORP), temperature (T), dissolved oxygen concentration (DO), and specific conductivity (SpC). For FC, the turbidity was measured in May 2019, December 2019 and July 2020 using a HI 93,414 portable turbidity meter (Hanna Instruments, S.L., Eibar, Spain) calibrated on site with fresh standards. The nutrient content of surface water in FC and LZ was analyzed in July 2020. Nitrogen as ammonium, nitrogen as nitrate, and phosphorus as phosphate were measured using Hach Lange cuvette tests LCK 304, LCK 339 and LCK 349 (Hach, Loveland, CO, USA), respectively, and a UV-VIS DR2800 spectrophotometer (Hach, Loveland, CO, USA).

Chemical composition of the water column at different depths was determined in FC (0, 10, 15, 25, 30, 45 m) and in LZ (0, 30, and 70 m) in October 2017. Samples were collected using a 5 L Van Dorn sampling bottle (KC Denmark). Water samples (125 mL) were filtered on site using a Merck-Millipore manual filtration unit and 0.45 µm nitrocellulose membrane filters (Merck Millipore, Burlington, MA, USA). Filtered water was stored in 125 mL polyethylene bottles, acidified with 1 M HCl and kept at 4 °C until chemical analysis in the laboratory. Suspended particulate matter (SPM) was sampled by additional filtering of 125 mL over the first 0.45 µm nitrocellulose membrane filter. The filter was washed by further filtration of 100 mL milliQ-water, and stored in separate plastic boxes after air-drying until chemical-mineralogical analysis by electron microscopy.

Samples for microbial community analysis and analysis of intact polar lipids (IPL) were obtained by filtration of biomass from the water samples over ~0.3 µm glass fiber filters (Advantec Grade G75 glass fiber filters, 47 mm, CA, USA). Prior to sampling, filters were heated at 350 °C for 30 min to remove residual organics. Water was filtered on site using a Merck-Millipore manual suction filtration system (Merck, Burlington MA, USA). The amount of water filtered per sample depended on the SPM content, which determined the volume after which filters became clogged. Filters were preserved on ice while sampling and stored at −20 °C until laboratory analysis. In FC, biomass samples were taken in October 2017 at 0, 15, 30, and 45 m depth, representative of the chemical zonation observed in the physicochemical profile at that time point, and detected in previous sampling campaigns [3,36]. In LZ, samples for microbial community analysis were taken from 0, 30, and 70 m depth, representing the mixolimnion and the upper and lower part of the monimolimnion.

### 2.3. Chemical Analyses and Electron Microscopy

Concentrations of element species in the water samples were measured by atomic absorption spectrometry (AAS) (Na, K, Ca, and Mg) using the Varian SpectrAA 220 FS, inductively coupled plasma-atomic emission spectrometry (ICP-AES) (Al, Cu, Fe, Mn, SO_4_, SiO_2_, and Zn) using an Agilent 7500ce, or inductively coupled plasma-mass spectrometry (ICP-MS) (As, Cd, Co, Cr, Ni, Pb, Se, Th, U) using a Varian Vista MPX instrument. The detection limit was 1 mg/L for major elements, and 1 μg/L for trace elements. 

SPM samples were examined with a JEOL JSM-7000F field emission scanning electron microscope (SEM) coupled with an Oxford INCA 350 energy-dispersive X-ray spectrometry (EDS) detector at the SGIker Facilities (UPV/EHU). The working conditions included an acceleration voltage of 20 kV, a beam current of 1 nA and 10 mm working distance. Samples were prepared by passing a small piece of carbon tape over the filter and adhered to the carbon holder. After a plasma cleaning process of 4 min it was coated with a 15 nm thick carbon layer in a Quorum Q150T ES turbo-pumped sputter coater.

Transmission electron microscopy (TEM) was completed at SGIker using a Philips CM200 microscope equipped with an EDS detector (EDAX Inc.) and LaB_6_ filament. An acceleration voltage of 200 kV was used. Selected area electron diffraction (SAED) impose substantial limitations on the particle thickness. Therefore, all samples were embedded in epoxy resin and cut at room temperature to 80 nm using a Leica EM UC6 ultramicrotome with a diamond knife, and mounted on 200 mesh Ni grids. Interplanar d-spacings were measured on SAED patterns with CrysTBox software [37] and identified using the American Mineralogist Crystal Structure Database [38].

### 2.4. Microbial Community Analysis

For DNA extractions, one filter per depth was cut into 6 pieces. Three were used for triplicate DNA extractions, with the exception of FC at 30 m, for which only 2 parts were successfully extracted. DNA extractions were performed using the FastDNA Spin Kit for Soil (MP Biomedicals, OH, USA) according to the manufacturer’s instructions. DNA concentrations were measured using a Qubit 2.0 fluorometer (Life Technologies, Darmstadt, Germany), with the Qubit dsDNA BR assay kit. PCR amplification of the V4-V5 region of the 16S rRNA gene was performed in triplicate per sample using the updated EMP primers [39]: 516F (5′-GTGYCAGCMGCCGCGGTAA-3′) and 806R (5′-GGACTACNVGGGTWTCTAAT-3′). The total reaction volume was 50 µL, consisting of 10 µL 5X HF buffer, 200 µM of dNTP, 10 µM of barcoded forward and reverse primers, 2 U/µL Phusion Hot Start II DNA polymerase (ThermoFisher Scientific), and 0.4 ng/µL DNA template. A negative control was included using nuclease-free water instead of template. After an initial denaturation step at 98 °C for 30 s, 28 cycles were run of 10 s denaturation at 98 °C, 10 s annealing at 56 °C, and 10 s elongation at 72 °C. A final elongation step was done for 7 min at 72 °C. Triplicate PCR reactions were pooled before clean-up and further processing. The pooled PCR products were cleaned with magnetic beads using the CleanNA PCR kit (GC Biotech BV, The Netherlands), and the final concentrations were determined using a Qubit 2.0 fluorometer (Life Technologies, Darmstadt, Germany). PCR amplicons were sequenced with the Illumina HiSeq 150 bp paired-end read sequencing (GATC Biotech, Konstanz, Germany). The sequences have been deposited in European Nucleotide Archive (ENA) at EMBL-EBI under accession number PRJEB38636 (secondary accession number ERP122073). Paired-end reads were processed using NG-Tax version 2.0 [40] on the Galaxy platform at https://www.systemsbiology.nl/ngtax/. The obtained 16S rRNA gene amplicon sequences were clustered into Exact Sequence Variants (ESVs) using an open reference approach. The following default settings were used: forward and reverse read length 70 nt, ratio ESV abundance 2, classify ratio 0.8, minimum percentage threshold 0.01, identity level 100%, error correction of 1 mismatch. In the paired-end libraries, only paired sequences were kept with perfectly matching barcodes. These barcodes were also used to demultiplex the reads by sample. ESVs were classified according to the SILVA SSU rRNA database (v128) [41,42]. ESVs were exported in the biom format and further analyzed with R (v 3.6.3) in RStudio (v 1.2.5042). Alpha diversity was calculated on rarefied data (sample size 14,600), and beta diversity was calculated on non-rarefied relative abundance data using the microbiome (v1.8.0) [43] and phyloseq (1.30.0) [44] packages in R. Graphs were plotted using ggpubr (v 0.2.5) [45]. The scripts used for data analysis can be found at https://github.com/mibwurrepo/van-der-Graaf-et-al-2020-Acidic-mine-pit-lakes.

### 2.5. Membrane Lipid Analysis

IPL extraction and analysis was carried out as described previously [46]. Briefly, freeze-dried glass fiber filters or freeze-dried biomass were extracted using a modified Bligh and Dyer procedure: ultrasonic extraction three times in methanol:dichloromethane:phosphate buffer (2:1:0.8, *v*/*v*/*v*) before the organic phase was separated by adjusting the solvent ratio to 1:1:0.9 and removed and dried under N_2_. Before analysis, the extract was redissolved in a mixture of hexane:2-propanol:water (72:27:1, *v*/*v*/*v*) and aliquots were filtered through 0.45 µm regenerated cellulose syringe filters. Analysis was carried out by high performance liquid chromatography/ion trap mass spectrometry (HPLC-ITMS) on an Agilent 1200 series LC coupled to an LTQ XL linear ion trap with Ion Max source with electrospray ionization (ESI) probe. All equipment and conditions were as described in [46]. The IPL groups were identified through comparison of their masses and fragmentation patterns with authentic standards or with those described in the literature [47,48,49]. IPLs were examined in terms of their peak area response and, due to ionization differences between the IPLs, may not reflect their actual relative abundance. However, this method allows for comparison between the samples analyzed in this study. For comparison purposes, IPL analysis was also performed on *Microbacter margulisiae* [50], which was taken from the culture collection at the laboratory of Microbiology at Wageningen University.

## 3. Results

### 3.1. Physicochemical Characteristics and Water Chemistry

#### 3.1.1. Filón Centro

The physicochemical profiles of the water column of FC obtained in our study and reported previously [3,36] showed that FC is a meromictic lake (Figure 2). An approximately 15 m thick mixolimnion is separated seasonally into an epilimnion, the upper layer that is regularly mixed due to heating by solar radiation (0–7 m), and a hypolimnion, where mixing is absent (7–13 m). A transitional redoxcline separates the mixolimnion from the lower, anoxic monimolimnion which extends to the lake bottom (50 m). The absence of mixing is partly illustrated by the temperature profiles of the water column (Figure 2a), which vary markedly in the epilimnion depending on the season: from 12 °C in December 2017 to 24 °C in October 2018. In the hypolimnion, the temperature variation is less pronounced, fluctuating between 10 and 12 °C, and below 15 m the temperature profiles overlapped regardless of the season. The small shift in temperature at 15 m and the gradual increase from 12 °C at 15 m to 16 °C at the lake bottom is likely the result of heat released by the exothermic oxidative dissolution of metal sulfides, mainly pyrite [51].

The depth and sharpness of the oxycline varies with the season (Figure 2b), with dissolved oxygen completely removed at 6 m (May 2007), 14 m (October 2017, October 2018) or 16 m (March 2008, December 2017, March 2019). This could reflect a variation in oxygen consumption by aerobic respiration between early summer, fall, and winter, as well as the temperature dependence of oxygen solubility. The redox potential (ORP) dropped from approximately +700 to +450–500 mV when oxygen was depleted (Figure 2c), and then dropped further between 25 and 30 m, from approximately +450 mV to between +300 and −100 mV, depending on the time of sampling. The pH profile showed a pronounced increase along the water column (Figure 2d). At the lake surface, the pH was highly acidic, between 1.9 and 2.4, and remained relatively constant down to a depth of 24 m. The pH increased sharply, from 2.5 to 3.9, between 24 and 29 m and then more gradually to approximately 4.8 between 40 m and the lake bottom. The specific conductivity (SpC) increased slightly between 15 and 16 m, from 5000 to 6000 µS/cm, and remained approximately constant until 27 m, after which it increased sharply to 13,000 µS/cm at the lake bottom (Figure 2e). 

Analysis of the chemical composition of the water column of FC showed that the concentrations of several major (Na, SiO_2_, Cu, Al, Zn) and trace (As, Be, Cd, Cr, Pb, Se, Th, Tl, U, V) elements decrease in the monimolimnion (Figure 3c,d, Appendix A). This indicates that they participate in (bio)geochemical reactions, which is referred to as reactive behavior. Most striking in these profiles is the increase in concentrations between the lake surface and the chemocline (28 m), followed by a sharp decrease towards the lake bottom, in the case of Cu, Al, Cr, Tl, Pb, Cd, Sb to close to 0 mg/L (Figure 3c and Appendix A). In the case of Zn, its reactive behavior was not as clear when considering only one time point, but it could be inferred from a decrease in concentration over time from 190 mg/L at 40 m in March 2008, to 151 mg/L at 45 m in October 2017 (Figure 3d). In contrast, the concentrations of several major (K, Mg, Ca, Mn, Fe, SO_4_^2−^), as well as trace elements (As, Co, Ni, Ag, Ba, Mo), remained constant or increased with depth (Figure 3a,b, Appendix A), indicating that they were not affected by (bio)geochemical reactions. This is referred to as conservative behavior. The total iron content increased from 824 mg/L at the lake surface, which was predominantly Fe^3+^ as reflected by the rusty red color of the water, to 7564 mg/L at 45 m (Figure 3a). The ORP profile indicated that below 18 m (ORP < +400 mV) (Figure 2c) iron was mostly present as Fe^2+^, which was supported by previous measurements of iron speciation along the water column [3]. The phosphate phosphorus (P-PO_4_^3–^) concentration in the mixolimnion (0 m) was 0.146 mg/L (Appendix A). Nitrogen as nitrate (N-NO_3_^–^) and nitrogen as ammonium (N-NH_4_^+^) were measured at 2.990 mg/L and 0.013 mg/L, respectively.

The turbidity in FC increased gradually from 2.7 to 7.0 nephelometric turbidity units (NTU) between 0 and 42 m depth, and then more sharply from 7.0 to 58–86 NTU (depending on the season) between 42 and 47.5 m, indicating a high concentration of SPM in this layer (Figure 2f). SEM-EDS analysis of SPM in the water column in the mixolimnion showed the presence of detrital phase and minor ferric oxides and (oxy)-hydroxysulfates (schwertmannite and jarosite). In contrast, SEM-EDS analysis of SPM from the deep monimolimnion of FC showed an increase in Cu content, from less than 0.5 weight percent (wt%) at 40 m, to 1.5 wt% at 43 m, coinciding with the first turbidity peak maximum (Figure 2f). At 43 m, rosette-like aggregates (0.33 µm) of lenticular Cu-S crystals were detected (Figure 4a), with a Cu to S molar ratio ([Cu/S]_m_) slightly above 2.0. This suggests the presence of reduced Cu minerals ranging from digenite (Cu_1.8_S) to chalcocite (Cu_2_S) or djurleite (Cu_1.96_S). At 45 m Cu accounted for 3.5 wt% in the SPM, showing up as globular Cu sulfides with [Cu/S]_m_ of 1.05–1.79, generally of 0.25–0.38 microns in size and with trace amounts of As. At 47 m, Cu sulfides ([Cu/S]_m_ = 1.3–2.2), and mineralized rods (Figure 4b) and spherical shapes resembling mineralized microbial cells became abundant, along with scarcer Zn-S spherical particles (Figure 4c). The Zn to S molar ratio in the ZnS particles was close to 1.0, which is coherent with the presence of wurtzite (ZnS). ZnS particles were generally larger in size than the Cu-S particles (0.66 µm against 0.25–0.5 µm for Cu), and both minerals, in some cases, showed incorporation of trace amounts of As and Cd. TEM-SAED analyses suggested the presence of digenite, djurleite and chalcocite (Figure 4d–f). Furthermore, TEM images and ring SAED patterns (typical of nanocrystalline samples) revealed that the apparently individual globular particles seen under SEM were actually aggregates of smaller discrete (nanometric) spherical crystals.

#### 3.1.2. La Zarza

Even though no time series data are available, the T and DO profiles (Figure 5a,b) indicate that the LZ pit lake is meromictic, with a very thin upper, oxic mixolimnion (0–4 m), and a lower, anoxic and reducing monimolimnion approximately 76 m thick. The physicochemical profiles obtained in October and December 2017 show a sharp drop in redox potential from +700 to + 450 mV in the upper 4 m, followed by a slight increase to +480–500 mV, after which the ORP decreases more gradually to +260 mV in the lowest 8 m of the lake (Figure 5c). The pH remained constant in the upper 8 m, approximately 2.0 (Figure 5d), and then increased to 2.5 between 8 and 16 m, followed by a slight decrease to 2.4 before increasing to 2.6 between 15 and 26 m. The pH remained constant until 70 m, and increased further from 2.6 to 3.0 towards the lake bottom at 80 m.

Analysis of the chemical composition of the water column in LZ indicated that several major (Ca, SiO_2_, Cu) and trace (Pb, Th) elements displayed clear reactive behavior, indicating their involvement in (bio)geochemical processes (Figure 3c and Appendix A). The specific conductivity in the water column increased sharply in the upper 10 m, from 12,000 µS/cm to 23,000 µS/cm, and remained constant until 45 m (Figure 5e). Between 45 m and the lake bottom (78 m), the SpC increased slowly to 26,000 µS/cm, suggesting that there was little precipitation of metals as insoluble compounds. SEM analysis of the SPM on the filter from 0 m showed the scarce presence of Fe(III) oxy-hydroxysulfates typical of oxidized AMD environments (schwertmannite and lesser jarosite) mixed up with detrital minerals, such as ore minerals (mainly pyrite and minor galena), phyllosilicates, quartz or rutile. Despite filtering the same volume of water as in the mixolimnion, very little SPM could be recovered from 30 m and 70 m, and no SEM images could be obtained of sulfide minerals at these depths. 

The concentrations of other dissolved major and trace elements increased with depth (Na, K, Mg, Mn, Fe, SO_4_, Al, Ni, As, Se, Be, Tl) (Figure 3e,f, Appendix A), indicating conservative behavior. This is in line with the initial increase and subsequent constant values determined for SpC, an indication of dissolved elements (Figure 5e). However, this parameter will mainly reflect the concentrations of Fe^2+^ and SO_4_^2−^ as they are an order of magnitude higher than the rest of the solutes (Figure 3e,f). The speciation of iron into ferrous and ferric iron was not determined in this lake, but from the clear, colorless water sample obtained at 30 m and using the ORP profile as a proxy for the Fe^2+^/Fe^3+^ ratio in these mine waters [52,53] it could be inferred that dissolved iron at that depth was mostly present as Fe^2+^. Due to practical difficulties, no depth profiles could be obtained for nitrogen and phosphate, and only the mixolimnion (0 m) was sampled in July 2020. The extremely high concentration of dissolved iron interfered with the colorimetric determination of nitrate, so that N-NO_3_^−^ could not be analyzed. The N-NH_4_^+^ and P-PO_4_^3−^ concentrations in the mixolimnion were 27 and 57 μg/L, respectively (Appendix A).

### 3.2. Microbial Diversity 

The microbial community composition in the water column of FC and LZ was investigated by 16S rRNA gene amplicon sequencing of filtered biomass from different depths. After the filtering and quality control steps, between 14,791 and 241,230 reads remained per replicate (Appendix A). A clear differentiation between the microbial communities was observed with depth, as reflected in the ordination plot (Figure 6). Most apparent was that in the mixolimnion of FC and LZ the communities were very similar, and that they differed strongly in the monimolimnion. Species richness (Chao1) increased towards the bottom of both lakes (Appendix A). It should be noted that the replicates for FC at 45 m and for LZ at 30 m and at 70 m showed a large variation, suggesting spatial effects of subsampling the filters. This is also reflected in the phylum- and family-level diversity observed in the individual replicates (Figure 7a). The overall diversity in FC increases with depth, as expressed by the Inverse Simpson index, which calculates a diversity index based on the total number of species and the relative abundance of each species. The McNaughton’s dominance (DMN), which measures the relative abundance of the two dominant taxa [54], was high in the mixolimnion of both FC (between 0.627 and 0.686), and LZ (between 0.612 and 0.647) (Appendix A). While in FC the DMN first decreased to 0.144 at 30 m, and then increased to 0.533–0.611 at 45 m, the DMN decreased progressively in LZ towards the lake bottom. 

#### 3.2.1. Filón Centro 

In the mixolimnion of FC (1 m), 79.0% of filtered reads were assigned to two genera, the autotrophic, aerobic, iron-oxidizing genus *Leptospirillum* (40.2% ± 7.5%) and the heterotrophic genus *Acidiphilium* (38.8 ± 9.1%), from the *Nitrospira* and *Alphaproteobacteria* class, respectively (Figure 7a,b). The family *Legionellaceae* accounted for 6.0 ± 0.4% of filtered reads, followed by sequences affiliated with three genera: *Metallibacterium* (1.9 ± 0.2%), *Acidibacter* (1.9 ± 0.3%) and *Acidocella* (1.4 ± 0.1%). The next sampling point was at 15 m, just below the oxycline, which at the time of sampling (October 2017) was situated at 13–14 m. A similar number of total ESVs (72) were detected at this depth compared to the mixolimnion (69) (Appendix A). Three genera accounted for an average of 68.4% of filtered reads: *Metallibacterium* (36.8 ± 1.9%), *Leptospirillum* (16.6 ± 0.8%) and *Acidithiobacillus* (15.0 ± 0.4%) (Figure 7b). In contrast to the upper layer, the euryarchaeal order *Thermoplasmatales* was detected (8.3 ± 0.8%): the genus *Thermoplasma* accounted for 2.4 ± 0.6% of total reads, and the family BSLdp125 for 4.7 ± 0.4%. One ESV, assigned to the *Chloroflexi* class JG37-AG-4, accounted for 5.8 ± 1.8% of filtered reads. The bacterial family *Acidobacteriaceae* (Subgroup I) accounted for 3.6 ± 0.7% of reads in this layer, with one ESV classified as an uncultured genus within *Acidobacteriaceae* (Subgroup I) accounting for 3.5 ± 0.7% of filtered reads (Figure 7b). 

At 30 m, the taxonomic diversity increased compared to the mixolimnion (1 m) and oxycline (15 m) (Appendix A, Figure 7a). Sequences classified as *Thermoplasmatales* accounted for 23.4 ± 1.9% of filtered reads, with the family BSLdp125 being the most abundant (8.1 ± 0.8%). Similar to 15 m, reads classified as an uncultured genus from the family *Acidobacteriaceae* accounted for 8.5 ± 2.2%. *Acetobacteraceae* accounted for 17.8 ± 1.1% of reads, of which an ESV representing an uncultured genus (8.5 ± 1.5%) and an ESV classified as *Acidiphilium* (7.6 ± 0.5%) were the most dominant ESVs in this ‘transition layer’. In addition, several taxa were detected at 30 m that are known to be involved in the microbial sulfur cycle: *Desulfurellaceae* (uncultured genus H16) (2.2 ± 0.3%), *Sulfuriferula* (1.4 ± 0.2%), *Desulfitobacterium* (0.9 ± 0.1%), *Desulfosporosinus* (0.4 ± 0.3%) and *Desulfomonile* (0.3 ± 0.1%). An ESV classified as unknown genus from the family *Peptococcaceae*, which also harbors the genus *Desulfosporosinus*, accounted for 7.9 ± 1.0% of filtered reads at 30 m.

In the lower, reducing monimolimnion (45 m), the microbial community showed a high number of ESVs, but 75.1 ± 5.0% of reads were represented by just four taxa (Figure 7a). Most striking is the dominance of *Desulfomonile*, accounting for 58.5 ± 3.5% of the total reads. An ESV classified as the sulfate-reducing genus *Desulfosporosinus* accounted for 2.0 ± 1.7% of total reads, and *Microbacter* and *Thermoanaerobaculum* for 8.3 ± 1.8% and 5.1 ± 2.1% of total reads. Sequences assigned to *Thermoplasmatales* were less abundant at 45 m than at 30 m, but still accounted for 2.7 ± 0.7% of reads. The detection of reads classified as methanogens, although in very low amounts and therefore not reflected in Figure 7, was of interest, as these are not commonly observed at such low pH. The euryarchaeal classes *Methanobacteria* (erroneously named bacteria), with *Methanobrevibacter* (0.018 ± 0.03%), and *Methanomicrobia*, with *Methanospirillum* (0.005 ± 0.007%), were detected.

#### 3.2.2. La Zarza

Similar to FC, the two dominant ESVs in the mixolimnion (0 m) of LZ were classified as *Leptospirillum* (67.6 ± 1.6%) and *Acidiphilium* (16.5 ± 2.5%) (Figure 7b). Reads classified as *Acidibacter* accounted for 2.0 ± 0.3%. As in FC, *Legionellaceae* was present in significant abundance: 7.4 ± 1.5% of filtered reads were assigned to two ESVs within this family. Because the lake surface and oxycline in LZ almost coincided or overlapped (Figure 5c), the oxycline was not sampled for DNA sequencing. Instead, the microbial community was sampled at 30 and 70 m, representing the upper and lower regions of the monimolimnion. At 30 m, reads assigned to the genus *Desulfocapsa* were the most abundant (16.4 ± 16.3%) (Figure 7a). The family *Desulfobulbaceae* (*Deltaproteobacteria*) accounted for 19.0 ± 18.9% of reads in this layer. The large standard deviation indicated a large spatial variation on the filter. The next most abundant genus was *Ralstonia* (11.1 ± 5.3%), followed by the euryarchaeal order *Thermoplasmatales* (17.0 ± 3.8%), with the family BSLdp125 (8.9 ± 1.5%) and genus *Thermoplasma* (5.6 ± 1.4%). The crenarchaeal genus *Acidianus* from the *Sulfolobaceae* accounted for 4.2 ± 1.4% of reads.

At 70 m, the lower region of the reducing monimolimnion, archaeal sequences represented 46.9 ± 16.6% of reads, compared to 22.2 ± 2.8% at 30 m and 0.5 ± 0.2% at 0 m (Figure 7a). Reads classified as *Thermoplasmatales* were most abundant, at 38.3 ± 22.8% of reads. Within *Thermoplasmatales*, the genus *Thermoplasma* accounted for 20.8 ± 12.5% of the total reads, and the family BSLdp125 for 14.4 ± 8.5%. *Acidianus* accounted for 6.1 ± 8.4% of reads. The large standard deviation in this layer again underscores the spatial variation on the filter. *Thaumarchaeota* accounted for 0.26 ± 0.12% of reads, with 0.21 ± 0.13% assigned to the class ‘*South African Gold Mine GP1*’. Bacterial sequences were dominated by *Acidibacillus* (13.4 ± 4.9%), *Ralstonia* (7.5 ± 8.2%), and *Desulfosporosinus* (4.4 ± 6.0%). Noteworthy is that, similar to the monimolimnion of FC (45 m) low amounts of sequences were classified as the euryarchaeal classes *Methanobacteria,* with *Methanobrevibacter* (0.012 ± 0.018%) and *Methanothermobacter* (0.006 ± 0.009%), and *Methanomicrobia*, with *Methanospirillum* (0.013 ± 0.018%).

### 3.3. Microbial Membrane Lipids

To assess the presence of unique microbial membrane lipid signatures in extreme AMD environments and their potential as biomarkers, the intact polar lipid (IPL) composition was determined in the filtered biomass from the water columns of FC and LZ. In the mixolimnion of LZ (0 m), IPLs were below the limit of detection. In the remaining filtered biomass sample extracts, a wide range of IPLs were detected (Table 1, including the isoprenoidal archaeal lipids glycerol dialkyl glycerol tetraethers (GDGTs) and archaeols (AR). The proportion of archaeal IPLs correlated positively with the relative abundance of archaeal 16S rRNA gene sequences in the different samples (linear regression, r = 0.98, *n* = 6; Figure 7a, Table 1). The non-archaeal IPLs included two types of aminolipids [ornithine lipids (OLs) and betaine lipids (BLs)], a wide range of phospholipids [phosphocholines (PC), phosphoethanolamine (PE), monomethyl PE (MMPE) and dimethyl PE (DMPE), phosphoglycerol (PG)] and glycolipids [sulfoquinovosyldiacylglycerol (SQDG), mono- and digalactosyldiacylglycerol (MGDG and DGDG)]. The core components of the phospholipids were either diacyl glycerol (DAG), diether glycerol (DEG), mixed acyl/ether glycerol (AEG) or ceramides (CR). Certain IPLs were detected uniquely or in high relative percent in FC 45 m. In particular, PC-AEG lipids accounted for 45% of the total in FC at 45 m, whereas they accounted for 14–23% at other depths in FC and in LZ 30 and 70 m. A series of ceramides with PE and MMPE headgroups were unique to FC 45 m (Table 1). The ceramide cores ranged in size between 32 and 36 carbons. For comparison purposes, we analyzed the IPL profile of *Microbacter margulisiae* (Appendix A), a *Bacteroidetes* species isolated from the nearby Tinto River and often found in co-occurrence with SRB in these environments [50].

## 4. Discussion

Depth profiles of pH and several metals in the water column of FC and LZ show an attenuation of the extreme acidity and toxic metal concentrations characteristic for AMD towards the lake bottom, which was most pronounced in FC. The upper, oxic layers (mixolimnion) of FC and LZ were similar and typical for acidic pit lakes [3], with a pH of approximately 2.0 (Figure 2d and Figure 5d) and ORP values of approximately +700 mV (Figure 2c and Figure 5c), dictated by the high concentrations of Fe^3+^. The dissolved metal concentrations in LZ were more extreme than in FC, however, with abnormally high concentrations of Fe (14,334 mg/L at 70 m depth), Al (up to 2178 mg/L), Mn^2+^ (up to 647 mg/L) and Zn^2+^ (up to 624 mg/L), and the trace elements As (up to 16,529 µg/L), Ni (up to 10,394 µg/L) and Co (up to 11,081 µg/L), to name only the most conspicuous ones (Figure 3 and Appendix A). This pattern of increasing pH and decreasing concentrations of certain metals is not always observed in (meromictic) acidic pit lakes in the IPB. For example, pit lakes such as San Telmo or Confesionarios show a stable monimolimnion with nearly constant pH and ORP values [3,15,36]. The physicochemical profiles for FC and LZ are similar to those of Cueva de la Mora pit lake (IPB) and Brunita pit lake outside of the IPB [4,30,55].

In both lakes, several dissolved metals, most notably Cu, were removed from the water column towards the lake bottom, indicating their involvement in (bio)geochemical processes affecting their solubility. In FC, Cu was removed completely at 45 m (Figure 3c). In LZ, the Cu concentration dropped from 184 mg/L at the lake surface to 23.2 at 70 m (Figure 3g). Although metal removal may be partly affected by other indirect mechanisms, such as adsorption to Al oxyhydroxysulfates [56,57,58], the detection of copper and zinc sulfides in the SPM from the monimolimnion of FC (Figure 4) strongly suggests the active coprecipitation of metals as metal sulfides. A similar metal removal mechanism was reported for the acidic IPB pit lake Cueva de la Mora [4,7] and more recently for the saline, acidic pit lake of the Brunita open pit mine in SE Spain [30]. In both cases, SRB were identified as the source of sulfide in the monimolimnion, which we hypothesized is also the case in FC and LZ. Microbial diversity analysis of the water column in FC and LZ supported this hypothesis, as it confirmed the presence of sulfidogenic taxa such as *Desulfomonile*, *Desulfosporosinus*, and *Desulfocapsa* in the monimolimnion.

In the mixolimnion of FC (1 m) and LZ (0 m) (Figure 6), the microbial community composition and physicochemical conditions were similar, and the community was dominated by *Leptospirillum* and *Acidiphilium* (Figure 7b). These two genera are typical for AMD environments and are mainly involved in the iron and carbon cycles, respectively [18]. *Leptospirillum* oxidizes Fe^2+^ to Fe^3+^, with O_2_ as electron acceptor, which is a key factor for the continuation of oxidative dissolution of metal sulfides by Fe^3+^ [59]. The high ORP associated with the Fe^2+^/Fe^3+^ couple (+710–+720 mV in sulfate rich acidic solutions) means that O_2_ is the only available electron acceptor for ferrous iron oxidation, restricting *Leptospirillum* to aerobic environments [18]. The strong oxycline in both lakes also supports high rates of O_2_ consumption. The *Leptospirillum* species described to date are obligate chemolithoautotrophic ferrous iron oxidizers, partly accounting for primary production in the lake [60,61]. Similar to our observations in Brunita pit lake, but contrary to what is generally seen in AMD waters, *Leptospirillum* outcompetes *Acidithiobacillus* as the dominant Fe^2+^-oxidizing genus [30,62]. This could reflect a lower tolerance of *Acidithiobacillus* to the extreme pH (approximately 2.0) and metal concentrations found in both lakes, favoring growth of *Leptospirillum* over *Acidithiobacillus* as reported previously [63,64].

The dominance of *Acidiphilium* in the mixolimnion of FC (1 m) and LZ (0 m) further underscores the importance of the iron and carbon cycles in this layer. *Acidiphilium* is capable of Fe^2+^ reduction, and some *Acidiphilium* spp. also carry out sulfur oxidation [65,66]. Except for *A. acidophilum*, which is capable of autotrophic growth [67], *Acidiphilium* species are obligate heterotrophs and play an important role in the removal of organic compounds such as organic acids that are toxic to autotrophic microorganisms like *Leptospirillum* [22]. Three genera found at lower abundance in the reads, *Metallibacterium* (1.9 ± 0.2%)*, Acidibacter* (1.9 ± 0.3%), and *Acidocella* (1.3 ± 0.3%) are also known for heterotrophic growth and likely play a similar role [68,69,70]. Interestingly, in both FC and LZ, 2 ESVs were assigned to the family *Legionellaceae,* one classified as the genus *Legionella* and the other as an uncultured genus. These were found at all depths investigated, with the highest relative read abundance in the upper layers, 6.0 ± 0.5% in FC (1 m) and 7.4 ± 1.8% in LZ (0 m). Although uncommon in AMD environments, the detection of reads classified as the family *Legionellaceae* was previously reported in acidic pit lakes [6,71,72], and could indicate a so far unexplored role for this taxa in AMD waters.

In the monimolimnion, the physicochemical profiles and microbial community composition of the two pit lakes diverged. The greatest difference was the more pronounced pH increase and metal removal, and the dominance of the SRB genus *Desulfomonile* in FC at 45 m (Figure 7b). In FC, the pH increased to 4.3 in the lower region of the monimolimnion (45 m) (Figure 2d), whereas in LZ the pH remained approximately 2.8 (Figure 5d). Concentration profiles of several dissolved elements, such as K, Fe, SO_4_^2−^ and Co, showed an increase in concentration towards the lake bottom. This is related to the conical shape of the pit lakes, which modifies the water/rock ratio and increases the amount of reactive rock surface per liter of acidic water. In the absence of (bio)geochemical processes removing dissolved elements from the water column, this decreased water/rock ratio will result in increasing concentrations with depth, referred to as conservative behavior. In contrast, reactive elements are removed from the water column towards the lake bottom, likely as insoluble mineral phases. Concentration profiles of dissolved elements in FC and LZ indicated a higher number of reactive elements disappearing from the water column in FC (Na, SiO_2_, Cu, Al, Zn, As, Be, Cd, Cr, Pb, Se, Th, Tl, U, V), than in LZ (Ca, Cu, Pb, Th, SiO_2_) (Figure 3, Appendix A). Since the oxidative dissolution rates of pyrite to Fe^3+^ and SO_4_^2−^ are orders of magnitude higher than for most other elements, removal of reactive elements from the water column is not clearly reflected in the specific conductivity (SpC) profiles. Furthermore, the inflow of water from underground mine tunnels with higher concentrations of dissolved elements cannot be excluded and would contribute to an increased SpC. The near-complete removal of Cu from the water column in FC, and the partial removal of Cu in LZ (Figure 3c,g) strongly indicates the formation of copper precipitates, most likely Cu_2_S and CuS as these are one of the first metal sulfides to precipitate according to their solubility products under these conditions [7]. The detection of zinc sulfides (ZnS) in the SPM in the lower layers of FC (Figure 4), and the turbidity peak observed at this depth (Figure 2f) indicate intense microbial activity and associated biosulfidogenesis. These observations strongly suggest metal removal through reaction with biogenic H_2_S, followed by precipitation as metal sulfides in the water column. 

Biosulfidogenesis in the monimolimnion of FC and LZ is supported by the detection of sulfidogenic taxa. This is most apparent in FC, where the monimolimnion is characterized by a transition from a diverse microbial community at 30 m, to a community at 45 m that is dominated by just one ESV (58.2 ± 3.5% of filtered reads) classified as the putative sulfate-reducing *Desulfomonile spp*. The two described *Desulfomonile* species, *D. tiedjei* [31] and *D. limimaris* [32], were isolated from neutrophilic 3-chlorobenzoate-reducing consortia originating from sewage sludge and marine sediments, respectively, and both strains are capable of autotrophic and heterotrophic sulfate reduction. Although no acidophilic *Desulfomonile* species have been described to date, this is not the first report of sequences classified as *Desulfomonile* in AMD environments. *Desulfomonile* was detected in the anoxic, reducing layer of the meromictic pit lake Cueva de la Mora [73], and more recently in the acidic Brunita pit lake [30], where it is also proposed to play a major role in the natural attenuation of extreme conditions in the monimolimnion. Its dominance in the water column, compared to more commonly found SRB species in AMD sediments such as *Desulfosporosinus* and *Candidatus* Desulfobacillus [26,27], suggests that this species is highly adapted to extreme acidity and metal toxicity, and provides an interesting candidate for application in bioremediation processes. 

In LZ, the attenuation of extreme acidity and metal concentrations was not as pronounced as in FC, but still a clear removal of Cu and Pb was observed (Figure 3c,g, Appendix A), usually the less soluble metals in AMD-affected zones [7,10]. Together with the detection of taxa known to perform sulfidogenesis from S_8_^0^, this supports biosulfidogenesis in the water column and the associated metal sulfide precipitation in the monimolimnion of LZ. Formation of S_8_^0^ in AMD environments occurs through the oxidative dissolution of certain metal sulfides (MS) through the polysulfide mechanism (equation 3) [59]. S_8_^0^ can then be oxidized aerobically or anaerobically to SO_4_^2−^ by sulfur-oxidizing microorganisms such as *Acidithiobacillus* spp., or (partly) reduced to H_2_S by sulfur-reducing or -disproportionating microorganisms [74]. In contrast to SO_4_^2−^ reduction, S_8_^0^ reduction is not H^+^ consuming, which fits the less pronounced pH increase in LZ.
(3)$$8\;MS+16Fe^{3+}\to 8M^{2+}+S_{8}^{0}+16Fe^{2+}$$

In the upper part of the monimolimnion (30 m) in LZ, one of the dominant ESVs was classified within the genus *Desulfocapsa*, of which two species have been described: *Desulfocapsa thiozymogenes* [75], isolated from a freshwater lake, and *Desulfocapsa sulfoexigens* [76], isolated from marine sediments. Both species are best known for the disproportionation of sulfur compounds such as S_8_^0^ or thiosulfate (S_2_O_3_^2−^) to H_2_S and SO_4_^2−^. As *D. thiozymogenes* and *D. sulfoexigens* are neutrophilic, and LZ is an extremely acidic environment with very high metal concentrations, the *Desulfocapsa* sequences detected in LZ likely represent novel (poly-) extremophilic species of this genus. Although *Desulfocapsa* was only detected in one of the two replicates, indicating the impact of spatial variation of the biomass on the filter, the absence of *Desulfocapsa* sequences in other samples supports the conclusion that this is not due to contamination and that this taxon plays a role at this depth.

In addition to the S_8_^0^ disproportionating genus *Desulfocapsa*, several S_8_^0^-reducing sulfidogenic taxa were detected at 30 m. *Acidianus* accounted for 4.2 ± 2.0% of filtered reads, and *Thermoplasma* for 5.6 ± 2.0%. The detection of *Acidianus* was surprising, since this genus is generally found in thermoacidophilic environments [77,78,79]. Isolated *Acidianus* spp. predominantly obtain energy through aerobic or anaerobic dissimilatory sulfur metabolism [80,81]. *Acidianus ambivalens*, for example, is capable of chemolithoautotrophic oxidation and reduction of S_8_^0^, and *Acidianus* strain DS80, is reportedly capable of both S_8_^0^ reduction and disproportionation [82]. Because the dissolved oxygen concentration in LZ at 30 m was below the detection limit, sequences classified as *Acidianus* likely indicate anaerobic respiration with oxidized sulfur compounds by this species, resulting in H_2_S production. Similarly, the detection of sequences classified as *Thermoplasma* (5.6 ± 2.0%), for which *Thermoplasma volcanium* and *Thermoplasma acidophilum* are the isolated representatives, suggests active H_2_S production. Both strains are capable of (heterotrophic) S_8_^0^ reduction, and were first isolated from coal refuse piles [83]. *Thermoplasma* belongs to the euryarchaeal order *Thermoplasmatales*, which is commonly detected in AMD environments under both aerobic and anaerobic conditions [84,85]. 

In the lower region of the monimolimnion in LZ (70 m), the removal of reactive elements was most pronounced, suggesting more active sulfidogenesis in this layer. Although not nearly as dominant as in FC, SRB were detected in this layer: sequences assigned to *Desulfosporosinus* accounted for 4.4% of filtered reads. At a pH of 2.9, this is one of the most acidic environments where this genus has been detected so far [26,27]. Although not uncommon in AMD environments, *Desulfosporosinus* is generally reported in AMD sediments where the potential for protection in less acidic micro-niches is higher, as mentioned above. The proportion of *Thermoplasma* increased (20.8%) compared to 30 m, as did *Acidianus* sequences (6.1%), further supporting active sulfidogenesis at 70 m. 

The detection of a low number of reads classified as methanogens in the water column of LZ and FC is noteworthy. Methanogenesis was previously observed in AMD environments [86], but so far only one acidophilic methanogen, *Methanoregula boonei* 6A8, has been reported, with an optimum pH of 5 and no activity below pH 4 [87]. An acidotolerant methanogen, *Methanobacterium espanolae GP9*, was isolated from sludge from a waste treatment facility of a pulp mill in Espanola (Canada), with an optimal pH range of 5.6–6.2 [88]. Lastly, the acidotolerant, hydrogenotrophic *Methanobrevibacter acididurans* ATM (OCM 804), isolated from an acidogenic digester, grows in a pH range of 5.0–7.5, with an optimum at pH 6.0 [89]. Acidophilic methanogens could be highly relevant for application in the treatment of organic waste, where acidification of the process conditions can pose a significant challenge [90].

We observed low concentrations of N-NH_4_^+^ and P-PO_4_^3−^, 13 and 146 μg/L in FC and 27 and 57 μg/L in LZ, respectively, and both lakes can be classified as oligotrophic [91]. For comparison purposes, in the eutrophic Brunita pit lake N-NH_4_^+^ and P-PO_4_^3−^ concentrations were 700 and 200 μg/L, respectively. The establishment of SRB in the water column of FC therefore shows that the potential for natural metal attenuation through biosulfidogenesis extends to even more extreme conditions than previously proposed from findings in the acidic pit lakes Cueva de la Mora and Brunita [4,30]. In the case of Brunita, an increase in pH and a removal of metals was observed in the lower monimolimnion, and together with the high abundance of SRB (*Desulfomonile*, *Desulfurispora*, *Desulfobacca*, and *Desulfosporosinus*), this pointed towards biosulfidogenesis in the water column. However, although the Brunita pit lake represents a worst case scenario in terms of salinity and metal concentrations, the presence of carbonate minerals in the open pit, not prevalent in the IPB [1,33], contributed to the reduction of acidity (driving pH to values of 5.0 in the pit lake bottom) and potentially aided in the establishment of SRB. Furthermore, the abundance of inorganic nitrogen (nitrate and ammonium) and phosphate in the water column provided ample opportunity for phototrophic primary producers to establish, which was also reflected in the observation of abundant green microalgae in surface waters and the high abundance of sequences classified as chloroplasts in the mixolimnion [30]. Growth of phototrophic microorganisms provided organic carbon to organotrophic microorganisms in the water column, which could aid the establishment of SRB. In Cueva de la Mora, the high availability of phosphorus enabled extensive phototrophic activity, thereby supplying organic carbon [4]. In contrast, the very low abundance of chloroplast sequences in FC and LZ and the absence of visual cues of algal growth in the mixolimnion, indicated very limited growth of phototrophic primary producers, and consequently a low availability of organic carbon in the two lakes. This further supports their oligotrophic nature. 

In addition to 16S rRNA amplicon sequencing, we characterized the IPL profiles for microbiological characterization of FC and LZ. The IPLs detected in the SPM from FC and LZ contained a range of amino-, phospho- and glyco-lipids of bacterial/eukaryotic origin as well as GDGTs and archaeols of archaeal origin. Overall, the distribution of IPLs was quite similar to that reported for a highly polluted meromictic lake in eastern Massachusetts [92]. In particular, there was, as per FC, a predominance of betaine lipids in the epilimnion. Also comparable between that lake, and FC and LZ was the increase with depth in diether PEs (PE-DEGs), which the authors of [92] attributed to SRB. Of particular relevance to our study were the ceramide lipids with PE and MMPE polar groups that were only detected in the FC 45 samples. PE ceramides are known to be dominant sphingolipids in several *Bacteroidetes* species [93], which corresponds well with the relative abundance of *Bacteroidetes* at this depth in FC. Among them was the *Bacteroidetes* genus *Microbacter*, accounting for 8.3 ± 1.8% of filtered reads. The only described species so far, *Microbacter margulisiae,* was isolated from acid rock drainage sediments in the nearby Tinto River [50]. *M. margulisiae* is a fermentative bacterium often found in co-occurrence with SRB in sulfidogenic enrichment cultures [14,50] and in natural and engineered acidic environments [9,28,94,95,96]. For comparison purposes, we determined the IPL profile of *M. margulisiae*, showing that PE ceramides account for approximately 15% of the total lipid sum in this species (Appendix A). As the *Bacteroidetes* species so far reported to contain PE ceramides (for example *Flectobacillus major, Bacteroides fragilis*, *Porphyromonas gingivalis*) are not typical for AMD environments [93,97], we propose that the detection of PE ceramides in these environments could serve as a biomarker for the presence of *Microbacter* species in future studies in similar systems.

Another interesting feature of the IPLs detected in the FC 45 m samples was the high proportion (45%) of phosphocholine lipids (PC) with mixed acyl/ether glycerol (AEG) core structures (Table 1). AEG lipids, which include plasmalogen lipids, are generally associated with anaerobic bacteria and are thought to contribute to membrane homeostasis in extreme environments [98]. PC-AEGs have been found as characteristic lipids of *Desulforhabdus amnigena,* which, like *Desulfomonile*, belongs to the order *Syntrophobacterales* [99], and we therefore postulate that the dominance of PC-AEGs at FC 45 m relates to the high proportion of the SRB genus *Desulfomonile* at this depth. 

Based on the results presented here, we propose that compared to Brunita and Cueva de la Mora, the FC and LZ pit lakes provide more extreme examples of natural (metal) attenuation mediated by biosulfidogenesis, with LZ on the far end of the spectrum, as its acidity and metal concentrations are among the highest found in the IPB and elsewhere [3,36]. We further propose that LZ is an example of this process in an earlier stage than FC, which is likely related to its age: LZ flooded in the 1990s, whereas FC flooded in the late 1960s. We suggest a scenario where initial attenuation in the form of metal removal is mostly mediated by biosulfidogenesis through reduction or disproportionation of S_8_^0^, triggering metal attenuation and enabling the establishment of SRB. SRB subsequently contribute to the amelioration of acidity, as sulfate reduction at low pH is H^+^ consuming. Contrary to what is often proposed [29], the availability of organic carbon does not seem to be the limiting factor, given that in both FC and LZ, phototrophic primary production is apparently very low. 

A better understanding of the biogeochemical processes that influence the development of these pit lakes, specifically the establishment of sulfidogenic microorganisms, could promote their function as low-cost natural remediation solutions for AMD waters. Acidic pit lakes furthermore pose extremely interesting targets for biomining of secondary resources, provided that the precipitated metal sulfides can be recovered efficiently. As an illustration of the economic potential of such lakes, a rough estimation of total copper removed from LZ pit lake through metal sulfide precipitation in the past 25 years indicates a removal of 400 tons, or approximately 16 tons of copper annually. In FC, this amounts to 8 tons Cu in total (over the past 50 years) or 160 kg/year, underscoring the extreme nature of LZ even by AMD standards. At the current market price of copper (~5800 US$/Ton), these rough estimates indicate a potential for biomining of mine water in acidic pit lakes via sulfidogenesis. In a global context of resource scarcity and the need for waste recycling, this option should be further explored.

## 5. Conclusions

This study shows that natural attenuation of extreme metal concentrations in acidic pit lakes through precipitation as metal sulfides is more widespread than previously recognized, and that biosulfidogenesis by sulfate-reducing and elemental sulfur-reducing and disproportionating microorganisms plays a key role in this process. The detection of SRB in the water column of the FC and LZ pit lakes is of great interest, as it expands the physicochemical limits under which SRB are observed to be active. FC and LZ are examples of natural bioremediation reactors at different stages of the process and at different degrees of extremophilic conditions, where S_8_^0^ precedes SO_4_^2−^ as the dominant inorganic sulfur compound for H_2_S production. Our results suggest that organic carbon availability is not necessarily the limiting factor for establishment of SRB, since both lakes could be classified as oligotrophic. The extreme acidity and metal concentrations suggest a high tolerance of the detected SRB species to these conditions, making them of great interest for the development of AMD bioremediation and biomining technologies.

## Figures and Tables

**Figure 1 microorganisms-08-01275-f001:**
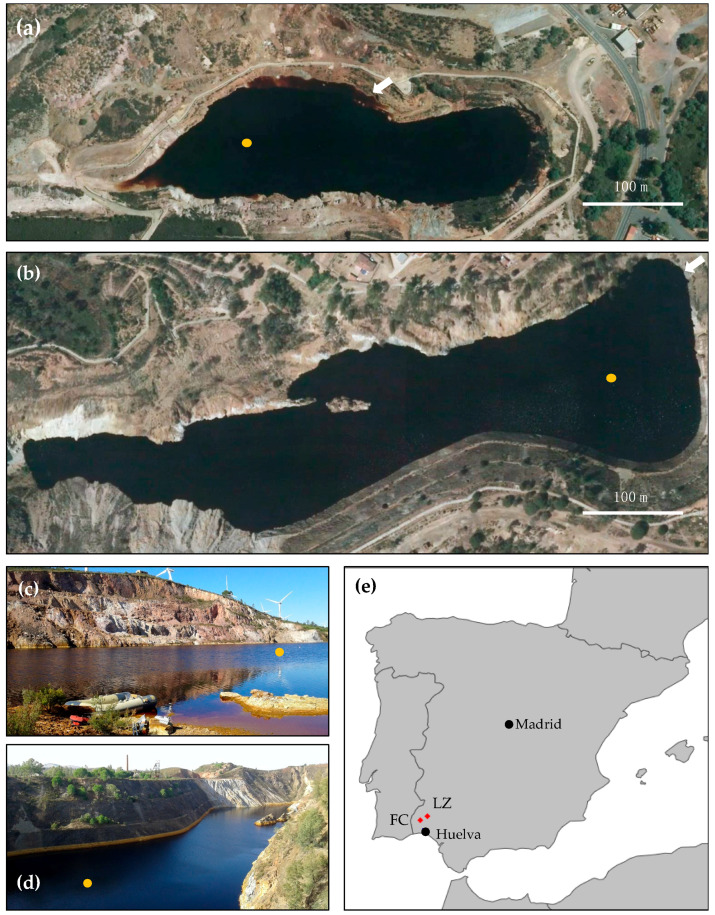
The acidic pit lakes Filón Centro (FC, Tharsis) (**a**,**c**) and La Zarza (LZ) at the La Zarza Perrunal mine (**b**,**d**); (**e**) the two pit lakes are located in Huelva province, southwest Spain. The orange dot indicates the sampling location. The white scalebar indicates 100 m. The arrows in (**a**,**b**) indicate the location and direction from which pictures (**c**,**d**) were taken, respectively.

**Figure 2 microorganisms-08-01275-f002:**
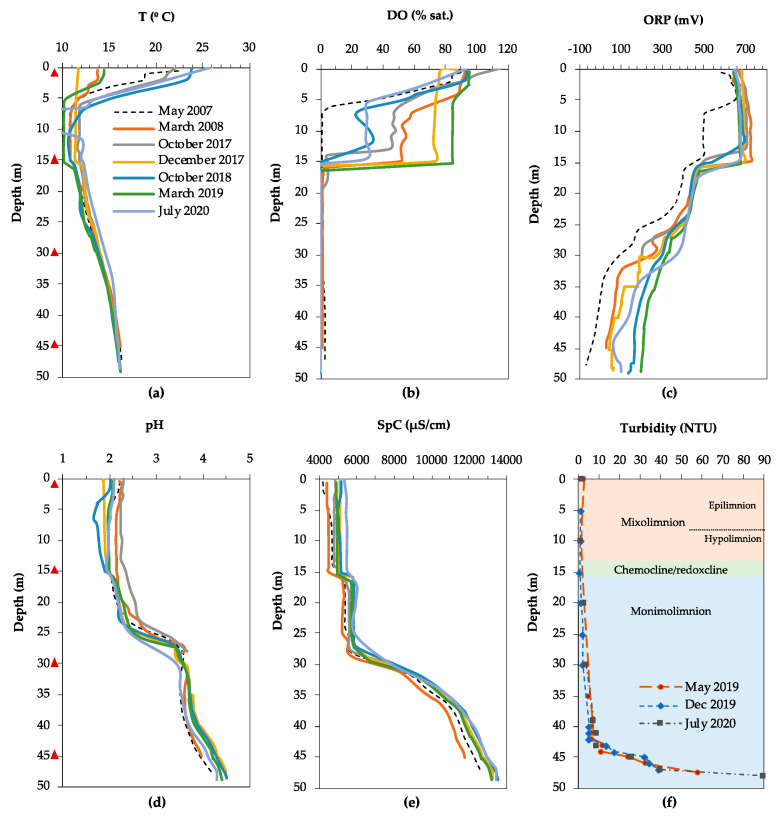
Physicochemical profiles of the water column in Filón Centro (FC). (**a**) Temperature (T), (**b**) dissolved oxygen (DO), (**c**) oxidation–reduction potential (ORP), (**d**) acidity (pH), (**e**) specific conductivity (SpC), and (**f**) turbidity (in nephelometric turbidity units (NTU)) profile combined with the chemical zonation of the water column inferred from the combined physicochemical profiles. Triangles indicate the sampling depths for biomass filtration in October 2017. Data from May 2007 and March 2008 were taken from previous studies for comparison purposes [3,36].

**Figure 3 microorganisms-08-01275-f003:**
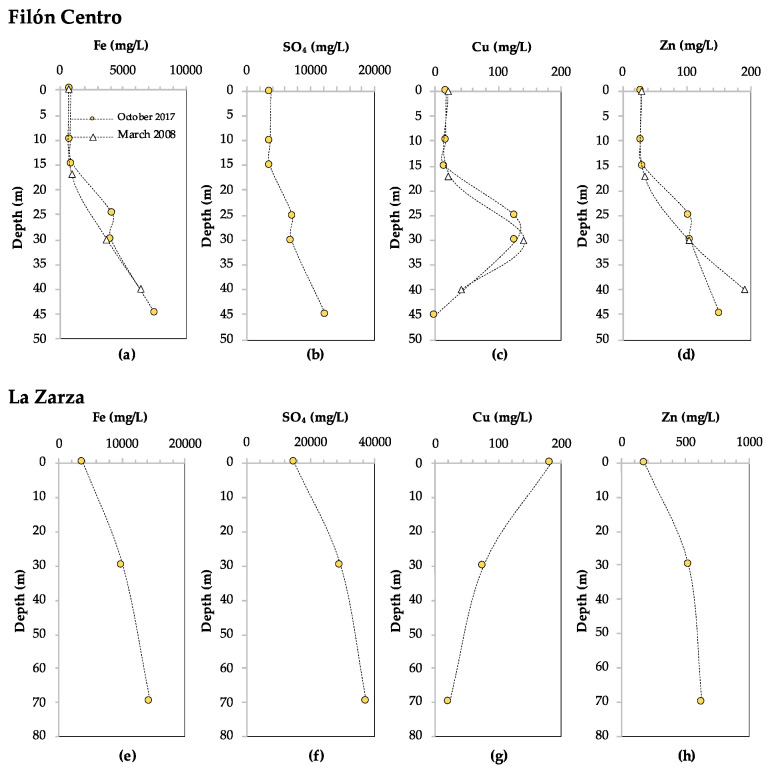
Depth profiles of total dissolved iron (Fe), sulfate (SO_4_^2−^), copper (Cu^2+^), and zinc (Zn^2+^) in mg/L in Filón Centro (**a**–**d**) and La Zarza (**e**–**h**), as measured in October 2017 (yellow circles). Data from FC are compared with chemical data from March 2008 [3,36].

**Figure 4 microorganisms-08-01275-f004:**
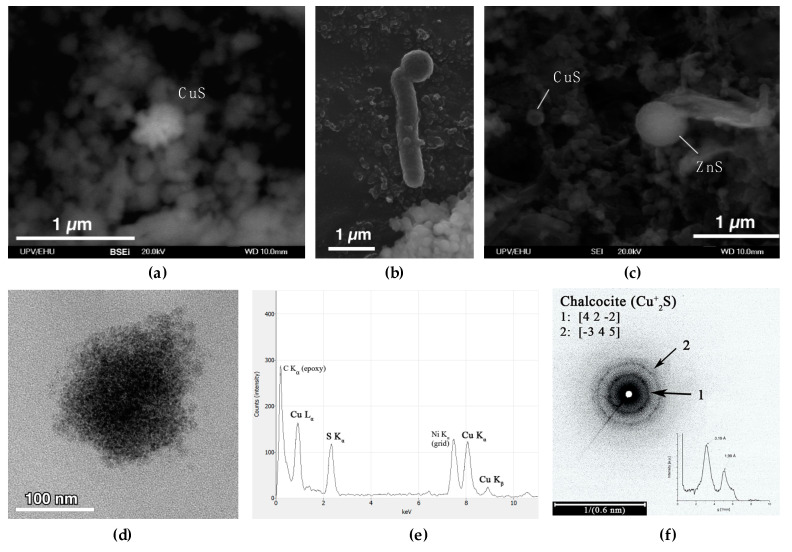
SEM (**a**–**c**) and TEM (**d**–**f**) images obtained on suspended particulate matter sampled at 43 m (**a**) and 47 m (**b**–**f**) depth in the acidic pit lake of Filón Centro in October 2018; (**a**) backscattered electron image of lenticular aggregate of chalcocite-like Cu sulfide; (**b**) mineralized bacterial rod associated with a spherical particle; (**c**) secondary electron image of globular ZnS with trace Cd and small darker particles which are globular Cu sulfides; (**d**) globular chalcocite (Cu_2_S) particle from SPM at 47m; (**e**) EDS pattern of the particle shown in (**d**) confirming composition dominated by Cu and S; (**f**) TEM selected area electron diffraction of globular sulfide shown in (**d**) with arrows indicating diffraction rings and calculated profile inset, characteristic of chalcocite (Cu_2_S) with 0.01 degree of confidence.

**Figure 5 microorganisms-08-01275-f005:**
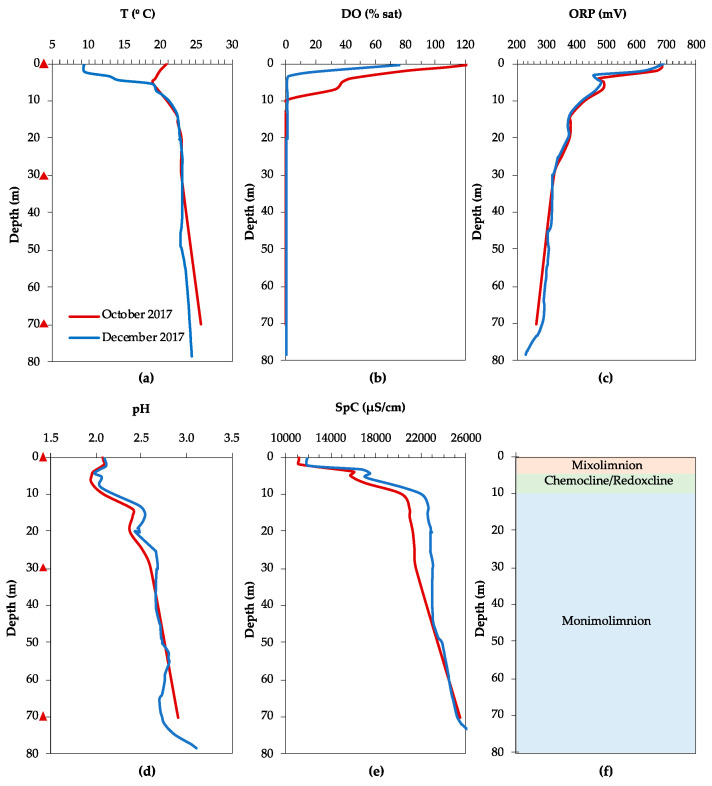
Physicochemical profiles of the water column in La Zarza. (**a**) Temperature (T), (**b**) dissolved oxygen (DO), (**c**) oxidation–reduction potential (ORP), (**d**) acidity (pH), (**e**) specific conductivity (SpC), and (**f**) inferred chemical zonation of the water column. Triangles indicate the sampling depths for biomass filtration in October 2017.

**Figure 6 microorganisms-08-01275-f006:**
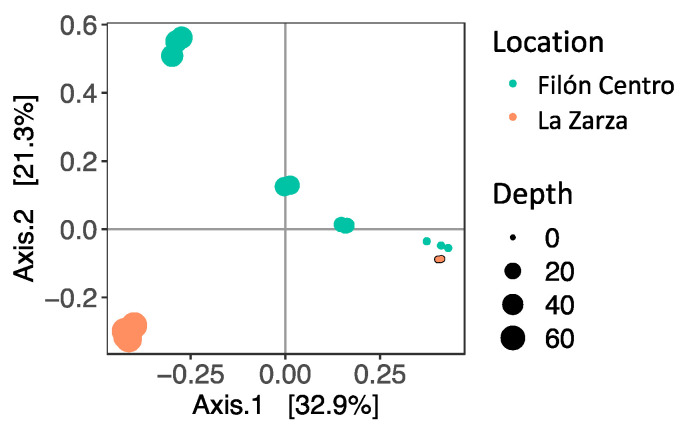
Ordination plot based on the Bray–Curtis dissimilarity index showing spatial separation between microbial community composition at different depths in Filón Centro (green) and La Zarza (orange).

**Figure 7 microorganisms-08-01275-f007:**
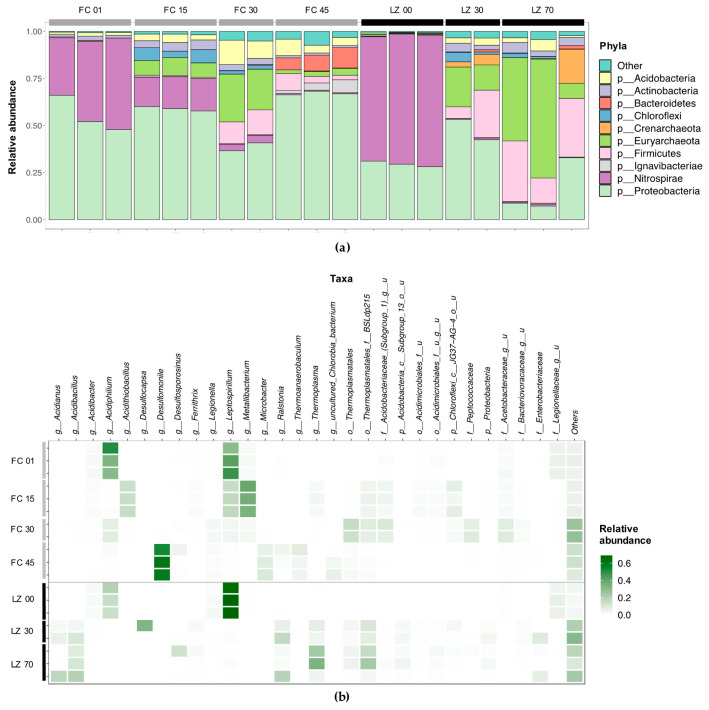
(**a**) Phylum-level composition in replicate samples of Filón Centro (FC, grey bars) and La Zarza (LZ, black bars) expressed as relative read abundance; (**b**) relative read abundance heatmap of the top 30 genera in individual replicates obtained per site, p_, c_, o_, f_, and g_ denote phylum-, class-, order-, family- and genus-level assignment of ESVs, respectively.

**Table 1 microorganisms-08-01275-t001:** Relative proportions of the intact polar lipid (IPL) groups, as determined by HPLC-ITMS. DAG: diacylglycerol, DEG: diether glycerol, AEG: mixed acyl/ether glycerol, CR: ceramides, PC: phosphocholine, PG: phosphoglycerol, PE: phosphoethanolamine, MMPE: monomethyl phosphoethanolamine, DMPE: dimethyl phosphoethanolamine, MGDG: monogalactosyldiacylglycerol, DGDG: monogalactosyldiacylglycerol, SQDG: sulfoquinovosyldiacylglycerol, BL: betaine lipid, OL: ornithine lipid, GDGT: glycerol dialkyl glycerol tetraethers, and AR: archaeol. The numbers in parentheses denote the number of different lipids within each group, differing by carbon number. Location is abbreviated as FC: Filón Centro, or LZ: La Zarza, with a number indicating depth; nd denotes not detected.

Polar Headgroup	PC			PG		PE				MMPE		DMPE	MGDG (6)	DGDG (4)	SQDG (3)	BL (18)	OL (8)	GDGT (14)	AR (4)
Core Structure	DAG (6)	AEG (42)	DEG (23)	DAG/AEG (3)	DEG (1)	DAG (10)	AEG (8)	DEG (6)	CR (5)	DAG/AEG (6)	CR (1)	DAG/AEG (5)							
Location																			
FC 1	1.1	19.6	7.9	5	0.2	6.5	1.8	nd	nd	nd	nd	7	0.1	1.7	4.7	36.1	0.6	nd	7.5
FC 15	1.3	14.4	3	1.7	0.4	16.4	5.8	nd	nd	11.7	nd	11.1	0.2	0.8	1.6	14	6.2	7.3	4.1
FC 30	2.4	21.5	2.4	1.4	1	10	5	nd	nd	6.3	nd	10.4	0.3	0.8	1.5	12.1	3	19.3	2.6
FC 45	0.9	44.6	1.4	0.7	1.2	3.9	1.9	3	4.3	3	0.7	4.6	0.4	0.9	2.5	12.9	1.9	9.3	1.8
LZ 30	16.7	23.2	0.2	0.3	0.6	5.9	6.8	2.7	nd	nd	nd	2.9	1.9	9.1	0	6.8	1	15.2	6.8
LZ 70	5.1	17.1	nd	nd	0.3	6.9	5.8	4.2	nd	nd	nd	3.7	2	12.8	0	9	nd	26.3	6.6

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
