# Peer review of "Biosulfidogenesis Mediates Natural Attenuation in Acidic Mine Pit Lakes"

_microorganisms, 2020, doi:10.3390/microorganisms8091275_

Round 1
Reviewer 1 Report
General comments:
This manuscript presents are significant amount of data and information. It will certainly be of interest to scientists conducting research on biogeochemical processes and bioremediation. However, as presented, it is a challenging read. Given the large amount of chemical and biological data provided, it would be helpful to the reader if the major finding for each section were clearly stated.
A concern of the data presented is that the water quality samples have no replicates. This is a significant limitation because the authors go to considerable length to connect the 16S data to the chemical conditions and biogeochemical explanations are presented based on gradients that rely on single samplings of the water. Without temporal and spatial replicates, the authors need to be more cautious with their conclusions.
Specific comments
50: awkward wording, perhaps start with: “Common characteristics of acidic pit lakes in the IPB are….”
84-85: awkward, reword
87-88: awkward, reword
90-91 awkward, reword
92: why is the use of () introduced here and not earlier, as it is not the first use of SRB?
104: awkward, reword
123: unsure of the meaning of “the mine was only exploited by underground works”, only as opposed to what?
128-129 redundant, reword
145-148: awkward, reword
234: awkward, reword
262: the meaning of conservative isn’t really defined until the end of the paper in the discussion, it is unclear here
344: comma after S before and
352: this transition into the presentation of the results could be smoothed out, introduce the idea a little before jumping in to where the methods left off
355: period instead of a comma at the end?
360: might want to define what Simpson evenness index is
362: awkward way to define McNaughton’s dominance
365: punctuation problem
Figure 7C: This font on the labels will be too small to be read
406: authors state that the finding is interesting, but don’t tell the reader why
419: state that the pH has increased, but never clearly provide reasoning
483: behavior defined as reactive, see comment for line 262
484-485: awkward wording and presentation of two separate findings
519: awkward, reword
542: punctuation, new sentence
543: punctuation
605-606: awkward, reword
607-609: awkward, reword
663: the timeline of these floodings seems important to the overall hypothesis, perhaps this point should be more clear in the introduction, it gets a bit lost in the results
669-671: good thoughts, awkward wording, reword
Reviewer 2 Report
An interesting investigation with good evidences of the main idea. But the article is rather bulky, since it presents an abundance of material. Thus, the main recommendation to the authors is to work on the effectiveness of the narration. It is necessary to select and emphasize the most important data which support the main idea, and possibly to delete the secondary data (or transfer to the supplamentary material).
For example, the sections about microbial membrane lipids in the Results (lines 439-467) and Discussion (lines 635-656) seem to be out of this idea. It represents some additional research which is poorly connected to the main purpose of the paper. It must be either justified better or excluded from the text.
Also the paper is negligently written. I have several comments and recommendations to improve it.
- All abbreviations should be decrypted at first mention. Please check it (for example, SPB).
- A lot of inaccurate links to figures and tables. Please check it throughout the text.
- Fig. 4b is not mentioned and discussed in the text. If it is not important it should be excluded from the text.
- Fig. 5. Only the data for samples from FC is presented. The data for LZ are discussed later (lines 341-345), but are not illustrated. It would be nice to make an illustration (separate one or added to the fig. 5)
- Fig. 6 b and c are mixed up. Look at the graphs and figure captions. It would be nice if this picture will be organized the same as fig. 2.
- Lines 574-576 "two strains", "both strains". The authors don't specify strains, they mention only two species. Thus "strains" should be changed to "species" or strain names should be specified.
- Line 674: "supplementary information XX". What is XX? There is no this information.
Reviewer 3 Report
A really excellent paper. Solid science and conclusions. Well-written, and a particularly good job on Figures and Tables. Most of my comments are simply editorial.
Pages are not numbered consecutively. After 15 it starts over again with 1, along with the logo.
In general, it is advised to always add a modifier when you use this or these. This what?
Good usage; with modifier as in 287, 477; lack of a modifier (and I may have missed some) 75 (or replace this with which), 100, 283, 358, 543, 546, 564, 631…
Throughout I think the use of Rio Tinto is better than Tinto River.
Introduction
40 change comma after (AMD), to a semi-colon
46 Change one of which being, to including
82 potentially protected [del somehow]
84 It is likely that the SRB…
101 why the use of ‘....’?
104-114 Excellent! Very clear.
Materials and method
Figure 1 is poorly placed; located 3 pages after the Fig is first mentioned in 131.
147 space; boxes after
153 Probably not important, but Burlington MA or NC?
159 …70 m depth,
172 how was coating done? Device used?
Results
Figure 1. I thinks these need a scale bar. And maybe an insert showing larger scale map (with scale and N arrow) to show location.
286 NTU; Define with first use. Nephelometric Turbidity Units (NTU)
362 Just as we always spell out OTU (Operational Taxonomic Unit), please define ESV with first use (Exact Sequence Variants).
Figure 7a; I can’t tell the differences between circles and triangles,
Figure 7 b. Is it possible in the text to break out the Proteobacteria by class? I expect a lot of Epsilon-. Always interesting. Discuss in 375.
455 Says ZP. LP?
Table 2. What is the number in parentheses? Number of variants? Is nd not detected or not determined? Add pit lakes by depth to caption?
Discussion
Starts with page 1, and has logo
[Amazing concentrations!]
567 Why use ‘…’
568 What is M in this equation? Just Metal?
625 no need to put worst case scenario in ‘…’
629 phosphate and phosphorous mean the same
640 Bacteriodetes in italics
663 age
667 Your standard for oligtrophic? Needs a reference.
674 Is there really a supplementary information XX?
708 Dr.
References
Not every work in a paper title gets capitalized; (Book chapter, yes): 6, 8.13, 30, 40, 53, 54, 61, 62, 80, 81, 96
- page numbers repeated
- del pp.
- is the year and the volume really both 2017?
- orebody repeated
- incomplete citation. Year in bold
51 Geochem
51 Issues
51 Soc.
68 Meth.?
81 volume? Correct doi?
83 odd embedded format commands
Table S2 Units?
Reviewer 4 Report
The interesting manuscript by Van der Graaf et al. investigated the natural attenuation of acidity and toxic metal concentrations in two meromictic and oligotrophic acidic mine pit lakes in the Iberian Pyrite Belt. Globally, they observed a significant removal of several dissolved metals and an increase in pH in the monimolimnion. Authors also have identified sulfidogenic microbial taxa were identified by using 16S rRNA metabarcoding and found that Desulfomonile was predominant.
This work is important for the bioremediation and biomining future research related to these peculiar environments.
The article is written in plain language and well-illustrated. Methods should be more detailed in the amplicon sequencing section (see comments below).
No major weakness is reported.
Below, some suggested edits.
Abstract
Line 29: add a comma after “knowledge”
Line 29: change “numbers” with “frequency”
Introduction
Line 43: “Earth”
Line 47: add a comma after “decade”
Line 54: remove “there”
Line 56: add a comma after “availability”
Line 60: change “just” with “only”
Line 65: add a comma after “Besides”
Line 100: change “this provided strong support for the functioning” with “this support the functioning”
Methods
Line 200: Please, add some more details for amplicon sequencing raw data processing. For example, did the authors removed the singletons? How they filtered the OTUs? Chimera filtering?
Discussion
Line 524: add comma after “monimolimnion”
Round 2
Reviewer 2 Report
Dear authors,
I carefuly read your response to my comments and I saw you have made a great job to improve a manuscript. I downloaded it with a good anticipation. But I saw the same file as the old version (before revision). I downloaded both versions and compared them. This is not my mistake - both files are similar! As I can judge a mistake has happen when you uploaded a wrong file.
So I put the same decision as before: Reconsider after major revision. Please upload the correct file.
Author Response
Dear Reviewer 2,
We do not know what went wrong, sorry for your inconvenience. When redownloading the resubmitted file on our end, we see the new one, but apparently this is not coming through. We will upload it again, this time also including highlighted in yellow some of the main structural changes.
Kind regards,
Lot
Round 3
Reviewer 2 Report
Dear Authors,
Nice job! The paper became much better and I'm satisfied with the quality of presentation now.